# Metabolic Disorders, the Microbiome as an Endocrine Organ, and Their Relations with Obesity: A Literature Review

**DOI:** 10.3390/jpm13111602

**Published:** 2023-11-13

**Authors:** Sorina Ispas, Liliana Ana Tuta, Mihaela Botnarciuc, Viorel Ispas, Sorana Staicovici, Sevigean Ali, Andreea Nelson-Twakor, Cristina Cojocaru, Alexandra Herlo, Adina Petcu

**Affiliations:** 1Department of Anatomy, Faculty of General Medicine, “Ovidius” University, 900470 Constanta, Romania; sorina.ispas@365.univ-ovidius.ro (S.I.); viorel.ispas@365.univ-ovidius.ro (V.I.); 2Department of Clinical Medicine, Faculty of General Medicine, “Ovidius” University, 900470 Constanta, Romania; 3Head of Nephrology Section, County Clinical Emergency Hospital of Constanta, 900591 Constanta, Romania; 4Department of Microbiology, Faculty of General Medicine, “Ovidius” University, 900470 Constanta, Romania; mihaela.botnarciuc@univ-ovidius.ro; 5Head of Blood Transfusions Section, County Clinical Emergency Hospital of Constanta, 900591 Constanta, Romania; 6Vascular Surgery Department, Cai Ferate Hospital, 35–37 I. C. Bratianu Boulevard, 900270 Constanta, Romania; 7Family Medicine, “Regina Maria” Polyclinic, 900189 Constanta, Romania; sorana.staicovici@reginamaria.ro; 8Department of Histology, Faculty of General Medicine, “Ovidius” University, 900470 Constanta, Romania; 9Preclinics Department II, Faculty of General Medicine, “Ovidius” University, 900470 Constanta, Romania; sevigean.ali@365.univ-ovidius.ro; 10County Clinical Emergency Hospital of Constanta, 900591 Constanta, Romania; 11Faculty of General Medicine, “Ovidius” University, 900470 Constanta, Romania; andreeanelsont@gmail.com; 12Family Medicine, 16 Cismelei Street, 900482 Constanta, Romania; cristyna_cojocaru@yahoo.com; 13Department XIII, Discipline of Infectious Diseases, “Victor Babes” University of Medicine and Pharmacy, 300041 Timisoara, Romania; alexandra.mocanu@umft.ro; 14Department of Mathematics, Biostatistics and Medical Informatics, Faculty of Pharmacy, “Ovidius” University, 900470 Constanta, Romania; adinapetcu@univ-ovidius.ro

**Keywords:** metabolic disorders, obesity, microbiota

## Abstract

The etiology of metabolic disorders, such as obesity, has been predominantly associated with the gut microbiota, which is acknowledged as an endocrine organ that plays a crucial role in modulating energy homeostasis and host immune responses. The presence of dysbiosis has the potential to impact the functioning of the intestinal barrier and the gut-associated lymphoid tissues by allowing the transit of bacterial structural components, such as lipopolysaccharides. This, in turn, may trigger inflammatory pathways and potentially lead to the onset of insulin resistance. Moreover, intestinal dysbiosis has the potential to modify the production of gastrointestinal peptides that are linked to the feeling of fullness, hence potentially leading to an increase in food consumption. In this literature review, we discuss current developments, such as the impact of the microbiota on lipid metabolism as well as the processes by which its changes led to the development of metabolic disorders. Several methods have been developed that could be used to modify the gut microbiota and undo metabolic abnormalities. Methods: After researching different databases, we examined the PubMed collection of articles and conducted a literature review. Results: After applying our exclusion and inclusion criteria, the initial search yielded 1345 articles. We further used various filters to narrow down our titles analysis and, to be specific to our study, selected the final ten studies, the results of which are included in the Results section. Conclusions: Through gut barrier integrity, insulin resistance, and other influencing factors, the gut microbiota impacts the host’s metabolism and obesity. Although the area of the gut microbiota and its relationship to obesity is still in its initial stages of research, it offers great promise for developing new therapeutic targets that may help prevent and cure obesity by restoring the gut microbiota to a healthy condition.

## 1. Introduction

According to the World Health Organization (WHO), the conditions of overweight and obesity are characterized by the presence of an atypical or excessive accumulation of adipose tissue, which may have detrimental effects on an individual’s well-being. Obesity in adults is defined by the organization as having a body mass index (BMI) equal to or more than 30, while overweight is defined as having a BMI equal to or greater than 25 [1].

Due to its universal applicability to individuals regardless of gender or age, BMI is the most effective indicator of overweight and obesity [2]. However, since this tool cannot compare the same level of fatness in various people, it should only be used as an essential reference. Age must be considered when defining overweight and obesity in children [3]. WHO defines it for those under five as weight-for-height greater than two standard deviations, and more than three standard deviations, respectively, over the WHO Child Growth Standards median [4]. For children between the ages of 5 and 19, obesity is defined as having a BMI higher than two standard deviations over the WHO Growth Reference median. At the same time, overweight is one standard deviation over [5].

According to World Obesity Atlas 2023, the annual economic cost of overweight and obesity will rise to USD 4.32 trillion by 2035 if preventive and treatment methods do not progress. As it can be seen in Table 1, by 2035, more than 4 billion people, which is more than half of the world’s population, will be overweight or obese if current trends continue [6].

The microbial community of the gut, which is involved in several physiological functions, is referred to as the gut microbiota (GM) [7]. It contains specific enzymes that can ferment some indigestible proteins and carbohydrates, which comprise around 10 to 30% of the energy consumed [8]. The primary byproducts of protein and carbohydrate fermentation are short-chain fatty acids (SCFAs), frequently called indirect nutrients. They supply around 10% of people’s daily energy needs, and in the colon, they are absorbed in a proportion of 95% [9].

GM both uses and produces micronutrients. Microbial species use an outside supply of nutrients and produce vitamins from scratch [9]. Most soluble B-vitamins, including cobalamin, thiamine, pyridoxine, biotin, folate, nicotinic acid, and pantothenic acid, as well as vitamin K2, can be produced by commensal bacteria species, such as Bacteroides, Enterococcus, and Bifidobacterium [10].

The two methods of the GM to acquire nutrients are external supply from other bacteria or de novo biosynthesis [11,12]. Small-molecule manufacturing is a high-energy operation, even though bacteria prefer to obtain these micronutrients from the environment whenever they are accessible. Studies on germ-free animals, who need extra B and K vitamins in their diets to stay healthy, show that the GM can synthesize them [13]. However, it is unclear how much the host’s systemic vitamin status is impacted by the vitamins produced in the gut. It should be emphasized that the host does not receive enough of the vitamins produced by the GM to meet its daily nutritional requirements [14]. Additionally, the bioavailability of microbially made vitamins for the host may be reduced by the inter-microbial exchange of micronutrients [15]. Nevertheless, it is worth noting that the GM can produce such vitamins.

Adults with healthy GM may adapt to various internal and external conditions [16]. Diet is a crucial factor that influences microbial diversity [17]. Intricate metabolic and transcriptional networks dependent on diet regulate the host–microbe and microbe–microbe interactions in the human GM [18,19]. A few instances of host–microbe interactions include symbiosis, in which both host and microbe species benefit; commensalism, in which one species benefits but the other is unaffected; and pathogenicity, in which one species benefits at the expense of the other [20]. The host or the bacteria can acquire nutrients that would otherwise be inaccessible thanks to symbiotic or commensal interactions [21]. Dysbiosis results from the mutualistic interactions between gut microbes, which leads to the loss of microbial diversity, the development of pathobionts, and disease [22].

Adipocytes, preadipocytes, fibroblasts, stromal cells, and macrophages comprise adipose tissue, a subclass of connective tissues [23]. According to the findings of this research, adipose tissue contributes to body homeostasis [24,25,26]. Adipose tissue secretes chemicals that regulate the body’s energy, lipid, and glucose metabolism, as well as immune system activities [27]. When fatty tissue no longer performs its hemostatic duties in an abnormal circumstance, such as obesity, this leads to the dysregulation of the mechanisms responsible for maintaining the stability of the internal environment and the activation of processes underlying the emergence of numerous metabolic disorders (MDs) [28]. Figure 1 illustrates the adipokine dysregulation pathway that might result in MDs and long-term problems during obesity [29].

A large study on 1760 female twins showed that visceral fat mass is significantly correlated with food and gut microbiota composition. Although fat mass growth is highly connected with total body weight, visceral fat mass has not been specifically affected by any weight loss methods currently available (such as hypocaloric diets, increased physical activity, appetite-suppressing medications, and bariatric surgery) [30]. It is essential to mention here that, in obese patients, visceral adipose tissue’s capacity to fight inflammation is influenced by the gut microbiota and its products [31].

Variability in the ability of adipose tissue to expand to store excess triglyceride quantities may impact metabolic issues [29].

Hypertrophic adipocytes can promote inflammation and heighten insulin resistance [32]. These cells release substantial amounts of pro-inflammatory cytokines like TNF, IL1, and IL6 [33]. Hypertrophic adipocytes encourage the emergence of diseases such as obesity-related insulin resistance [29]. Small adipocytes have anti-inflammatory capabilities and increase glucose absorption in insulin-sensitive tissues.

Furthermore, it has been demonstrated that the size of the adipocytes, rather than their amount, is associated with the risk of diseases linked to nutrition [34]. Adipose tissue mass increases when there is an imbalance between the energy obtained from meals and that used for metabolism and physical activity. Metabolism is also impacted by adipokines that are overly produced by adipose tissue [35].

## 2. Materials and Methods

Using the keywords “metabolic disorders and obesity” and “microbiome and obesity”, the literature on PubMed was searched for relevant studies. We conducted a comprehensive manual search to identify all relevant original papers. This included using references from top search results, reviews, and other scholarly publications. As this study is a literature review, it is important to note that ethical approval is not necessary.

The inclusion criteria only included randomized clinical trials (RCTs) and papers that were published between the timeframe from 1 January 2017 to 31 December 2022 and that were freely available in English with full-text access.

The exclusion criteria included two factors: sample size, where only publications with a minimum of 100 participants were considered, and peer-reviewed literature, wherein research without peer review was deemed ineligible for inclusion. Experiments that lacked full data and those that failed to provide measurable outcomes were also excluded from the analysis.

The Patient, Intervention, Comparison, and Outcome (PICO) framework was used as the foundational structure for a systematic review, serving as a methodological approach.

Population: overweight and obese individuals randomly selected for control trials.

Intervention: those selected for the study either followed a carefully designed weight plan diet or a placebo diet.

Comparison: changes in various markers were assessed, as well as the level of adiposity in the weight-loss diets.

Outcomes: to see if there is a connection between gut microbiota and obesity.

The review was reported using the Preferred Reporting Items for Systematic Review and Meta-Analysis (PRISMA) guidelines.

The studies that examined the relationship between metabolic disorders, microbiota, and obesity were carefully chosen for further examination. Positive relationships (noted with “YES”) and negative associations (registered with “NO”) between metabolic disorders, microbiota, and obesity were divided into two groups for this review. “YES,” correlations indicated evidence linking the three topics.

After scanning the PubMed database, 1345 citations were generated (Figure 2). After removing 253 duplicate items, the list still included 1067 articles. Additionally, 204 articles that did not meet the search criteria but were accidentally included in the results were subsequently excluded. Among the total number of studies considered, a subset of 305 studies were excluded due to a clear mismatch with the predetermined criteria outlined for our research. Additionally, 104 papers were further eliminated as they failed to directly address the specific question of interest. Furthermore, an additional 35 studies were disregarded due to the unavailability of complete text access. Moreover, 164 studies were excluded as they were primarily centered on an age group that did not align with the intended focus of our investigation. Lastly, one article was unintentionally overlooked due to being written in a language other than English. At the present moment, there exists a total of 279 search results that meet the criteria for our inquiry.

A total of 10 publications that met the criteria for inclusion were therefore included in the final literature evaluation. The data derived from the aforementioned studies is shown in Table 2, which is provided inside the Results section.

The following is a comprehensive analysis of the findings obtained from the PubMed search, including the specific criteria used over the course of our inquiry.

The search for “metabolic disorders and obesity” yielded a total of 1184 items.

The search for “microbiome and obesity” yielded a total of 161 search results.

The search parameters used in our inquiry on PubMed included the following filters: availability of unrestricted complete texts, inclusion of randomized controlled trials, involvement of human subjects, and publication dates spanning from 1 January 2017 to 31 December 2022.

All relevant information was extracted and inserted into an Excel spreadsheet.

## 3. Results

The studies analyzed in Table 2 show that the gut microbiota has an influence on the host’s metabolism and obesity via several mechanisms, including gut barrier integrity and insulin resistance, among other variables [36,37,39,40].

While the investigation into the gut microbiota and its connection to obesity is still ongoing, it has considerable potential for the identification of novel therapeutic targets. By returning the gut microbiota to a state of optimal health, it may be possible to mitigate and treat obesity effectively.

Consequently, our results show that individuals with varying medical conditions who are classified as obese, discovered features of gut microbiota, and established correlations between bacterial commensals and other clinical manifestations [39]. The aforementioned findings have identified potential targets for adjuvant therapy in the management of obesity accompanied by metabolic abnormalities. Additionally, these findings have shed light on the involvement of gut microbiota in the development of metabolic diseases [43,44,45].

The selected studies also show that the gut microbiota has the potential to serve as a predictive indicator for assessing the impact of a prebiotic intervention on the mood of individuals with obesity [38,39]. Identifying essential gut bacteria involved in the body’s response to food-based treatment might facilitate the customization of these techniques. Based on our findings it has been suggested that neuroactive properties and might potentially serve as a reliable biomarker for assessing the response of gut microbiota to prebiotics [39,40,41].

## 4. Discussion

Obesity is associated with a multitude of comorbidities spread across several different organs.

Figure 3 shows that the comorbidities can be classified into three major classes: metabolic, like cardiovascular diseases, type 2 diabetes, nonalcoholic fatty liver disease (NAFLD), breast, colorectal, endometrial, esophageal, kidney, ovarian, pancreatic, prostate, and gastro-esophageal reflux disease, as well as heart failure with preserved ejection fraction; mechanical, such as asthma, chronic back pain, and knee osteoarthritis; and mental, such as depression and anxiety [46].

While the etiology of obesity has typically been attributed to a calorie surplus compared to calorie expenditure, recent research has linked the illness to genetics and MB. Still, recommended as the initial step in weight loss is lifestyle improvement, which includes food intake changes [47].

### 4.1. Hormones Involved in Visceral Obesity and Microbiota

The functional interaction between gut microbial products and the host endocrine system has also been demonstrated to indirectly change the conventional hormonal responses to cortisol, ghrelin, leptin, glucagon-like peptide 1, and YY [48].

Different regions within the gastrointestinal tract regulate energy intake and glucose homeostasis through the release of small molecules: the stomach releases ghrelin, the duodenum releases cholecystokinin, gastric inhibitory peptide, and ghrelin, the jejunum releases GLP-1, peptide YY, and gastric inhibitory peptide, and the ileum releases GLP-1, oxyntomodulin, peptide YY, and FGF-19. The colon releases GLP-1, oxyntomodulin, and peptide YY. The pancreas produces insulin, glucagon, amylin, and pancreatic polypeptide [49].

As it can be seen from Figure 4, the signal release is influenced by nutrient intake, bile acids, and the gut microbiome [50].

### 4.2. Neurotransmitters and Neuropeptides Ivolved in Obesity

Amino acids, monoamines, trace amines, peptides, gasotransmitters, purines, and smaller substances like Ach and anandamide are neurotransmitters. The most significant neurotransmitters are Glu, GABA, glycine, DA, NE, 5-HT, and histamine [18].

The colon is mostly inhabited by Bifidobacterium, Lactobacillus, Lachnospiraceae, Blautia, Coprococcus, Roseburia, and Faecalibacterium, which are responsible for the synthesis of the primary short-chain fatty acids (SCFAs) that serve as an energy source for epithelial cells. These SCFAs include butyrate, acetate, lactate, and propionate [51]. In individuals with inflammatory bowel disease, there is an observed decrease in fecal short-chain fatty acid concentrations, as well as a decline in the abundance of Firmicutes and Bacteroidetes [52].

Butyrate induces apoptosis in colon cancer cells and has a vital role in facilitating oxygen absorption by epithelial cells, while also mitigating inflammatory responses [53].

Short-chain fatty acids regulate the differentiation of T-cells within the context of immunological cells. They have the ability to induce the secretion of gastrointestinal hormones inside enteroendocrine cells, and also have a significant impact on the regulation of neuronal pathways and central nervous system signaling [54].

Based on the extensive body of research, it becomes apparent that SCFAs synthesized by microorganisms play a crucial role in facilitating communication within the microbiota–gut–brain axis, safeguarding the integrity of the intestinal barrier, and modulating inflammatory reactions [55]. The brain is exposed to short-chain fatty acids via the circulatory system, as they are transported from the gut microbiota. Figure 5 shows how these SCFAs have the ability to modulate the functioning of astrocytes, microglia, and neurons inside the brain [56].

### 4.3. Microbiota

Less than 10% of the DNA in the human meta-organism is thought to have Homo sapiens origin [57]. According to data derived from the HMP and the MetaHIT collaboration, the human gastrointestinal tract holds a total of 2766 distinct microbial species [58]. The gut microbiota is mostly composed of Proteobacteria, Firmicutes, Actinobacteria, and Bacteroidetes bacteria, which together account for over 90% of its composition. The Firmicutes bacteria are the predominant microorganisms found in the GI tract, with Lactobacillus species and Gram-negative Bacteroidetes being the most prevalent within this taxonomic category. Fusobacteria and verrucomicrobia comprise the remaining 10% of the gut microbiota.

Figure 6 shows how the microbiome is influenced by several factors, including hormonal fluctuations, changes in food patterns, heightened stress levels, and physiological modifications resulting from the administration of medications, notably antibiotics [59].

### 4.4. Gut–Brain Axis

The gut–brain axis refers to a bidirectional communication pathway between the gut microbiota and the brain, which is implicated in processes such as aging, neuronal development, and brain function [60].

The branches of the vagus nerve that are connected to the gastrointestinal tract have a role in regulating the secretion of glandular tissue by modulating the contraction and relaxation of smooth muscles. The duodenum and the rest of the gastrointestinal tract are anatomically linked to the celiac trunk, a major branch of the vagus nerve, located near the distal portion of the descending colon. The lamina propria and muscularis externa layers establish connections with the preganglionic vagal nerve (neurons located in the medulla. The nucleus tractus solitarii (NTS) serves as the origin of neural impulses that are sent to various brain regions, including the locus coeruleus, amygdala, thalamus, and rostral ventrolateral medulla. These signals are received by the NTS from sensory cells located in the nodose ganglia [61].

It has been suggested that metabolites, including SCFAs and neurotransmitters, have an impact on the levels of related metabolites in the brain through blood circulation [62]. This influence plays a role in regulating brain functions and cognition, in addition to the well-established hypothalamic–pituitary–adrenal axis and endocrine pathways (specifically, intestinal peptides and hormones) [63]. The gut microbiota has the potential to enhance the transmission of information to the brain via its influence on the local neurological system. The peripheral immune system may be activated by lipopolysaccharide and other endotoxins synthesized by bacteria. This activation prompts the migration of peripheral immune cells into the brain, resulting in inflammation inside the central nervous system. This inflammatory response includes immune cell activation and the creation of cytokines, among other processes [64].

Figure 7 shows that, apart from serving as precursors for neurotransmitters, gut bacteria has the ability to facilitate the production of neurotransmitters via the metabolic breakdown of food [63]. The upregulation of the rate-limiting gene TPH1 in enterochromaffin cells allows for the modulation of serotonin production via the utilization of metabolites produced by spore-forming bacteria, which serve as signaling molecules. They also stimulate enteroendocrine cells to synthesize neurotransmitters, which can also be synthesized by bacteria and enteroendocrine cells, thus facilitating their distribution throughout the body via the circulatory system [65]. Certain neurotransmitter precursors have the ability to traverse the BBB, enabling their involvement in the brain’s manufacturing cycle of neurotransmitters. Moreover, GLU serves as a neurotransmitter synthesized and secreted by neuropod cells located in the intestinal epithelium. The modulation of neurotransmitter synthesis by the gut microbiota has the potential to impact cognitive performance in neurological disorders such as Alzheimer’s disease, Parkinson’s disease, autism, and schizophrenia [63].

### 4.5. Gut Microbiota and Neurotransmitters

Certain neurotransmitters are involved in the interaction between the gut microbiota and the host, and they are influenced by the control of their precursors [63]. Additionally, it explores the role of intestinal enteroendocrine cells in the synthesis and release of neurotransmitters, which are facilitated by microbial activity [18].

Table 3 presents a comprehensive overview of the synthesis of neurotransmitters and their respective functions within the gut–brain axis, which are under the regulation of the GM [63].

Recent research has shown that the metabolites produced by the gut microbiota include not only short-chain fatty acids, bile acids, and histamine, but also include distinct chemical messengers such as glutamate, gamma-aminobutyric acid, dopamine, and serotonin [18].

The communication pathway between the brain-generated signals and entero-epithelial cells is referred to as the hypothalamus pituitary adrenal axis. SCFAs generated by the gut microbiota have the ability to interact with neurons or penetrate the circulatory system. The composition of gut microbiota has been shown to have an influence on the processes of neurotransmitter production and degradation [18].

Through the process of gene encoding, several bacteria possess the ability to produce distinct enzymes that facilitate the conversion of substrates into essential neurotransmitters [66]. Certain metabolites produced by bacteria have the potential to function as signaling molecules, therefore triggering the synthesis and secretion of neurotransmitters by enteroendocrine cells.

According to recent research [67,68], it has been shown that some strains of Lactobacillus have the potential to induce the release of acetylcholine. The major neurotransmitter of the parasympathetic nervous system is responsible for several physiological functions, including the contraction of smooth muscles, dilatation of blood vessels, secretion of body fluids, and control of the heart rate [69,70].

In order to maintain homeostasis, the regulation of food intake involves the interaction between orexigenic and anorexigenic signals that are created in the stomach and conveyed by the vagus nerve, ultimately influencing certain areas of the hypothalamus [71]. The interaction between gut bacteria and food-related variables is known to influence the regulation of orexigenic and anorexigenic peptides released by enteroendocrine cells in the distal small intestine, with the microbial byproducts of the gut playing a significant role in this process. The balance of orexigenic and anorexigenic signals in the hypothalamus is modified as a consequence [72]. Moreover, the brain–gut interaction may be shown by the presence of neuroactive metabolites such as lipopolysaccharides and tryptophan metabolites, which are produced by gut microbes. Several brain networks, including the prefrontal cortex, the dopaminergic reward system, and the sensorimotor system, engage in interactions to regulate the hedonic aspects of food consumption. The activation of the extended reward system may occur via exposure to food ads and other environmental signals, hence superseding the regulatory mechanisms of homeostasis [73].

### 4.6. Incretin Effect

Peptide hormones GLP-1 and GIP play a significant role in postprandial metabolism. They are produced in response to nutrient consumption by enteroendocrine cells in the colon [73]. The incretin effect, their most advantageous condition, enhances the glucose-stimulated insulin release from the pancreas by discovering glucose equilibrium. Though they synergize when given together, GIP is thought to be the primary incretin hormone responsible for this effect [74].

The efficiency of the endogenous GLP-1 and GIP is increased by the dipeptidyl peptidase-4 (DPP-4) inhibitors by impeding the rapid DPP-4-mediated breakdown of these molecules [75]. While this is happening, the GLP-1 receptor agonists, which profit from a change in structure, increase DPP-4 resistance by hastening the beginning of the GLP-1 receptor. In addition to decreasing pancreatic glucagon release via alpha cells, GLP-1 RAs also speed up stomach emptying, reduce appetite, and reduce nutritional intake, all of which contribute to weight reduction in a way different from glucose-dependent stimulation [76].

Gut microorganisms use different strategies to communicate with host cells. Short-chain fatty acids are byproducts of microbial fermentation of various nutrients; these SCFAs are detected by specific G-protein-coupled receptors that are expressed on the surface of enteroendocrine cells such as L-cells and result in the production of GLP-1, GLP-2, and PYY. Additionally, indoles, which are bacterial byproducts of tryptophan breakdown, regulate GLP-1 secretion and hunger [77].

Therefore, it seems that GLP-1 has an affinity for receptors located in the gastrointestinal tract (GIT), leading to the transmission of signals to the brain via sensory neurons. The release of insulin, which is triggered by GLP-1, and the suppression of glucagon production work together to maintain low levels of glucose in the bloodstream during or shortly after a meal. GLP-1 exerts an additional physiological effect known as the ileal brake, which manifests as a deceleration of gastrointestinal motility [78].

### 4.7. GIP

The secretion of gastric inhibitory polypeptide from enteroendocrine K-cells, which is stimulated by butyrate, is a powerful stimulator of insulin secretion that is dependent on glucose levels. This action of GIP works in conjunction with GLP-1 [59].

Growing evidence suggests a close interaction between GIP and the gut microbiota. Some studies have shown that gut microbiota composition can influence the release and activity of GIP [79]. For example, certain bacterial species in the gut have been found to promote the secretion of GIP, while others may inhibit it. Conversely, GIP has been shown to impact the gut microbiota. It can affect the composition and function of gut bacteria, potentially influencing their growth and metabolism [80].

## 5. Conclusions

Obesity is rapidly increasing in prevalence, which is turning it into a serious public health issue. Distinct geopolitical situations have particular exposure considerations. The main factors are sedentary lifestyles, urbanization, migration from rural to urban areas, consumption of energy-dense meals, and lack of physical activity. There are many potential biomarkers for identifying obesity, including microRNAs, adipocytes, oxidative stress, and microbiota. Given the magnitude and interconnectedness of the effects of obesity, a comprehensive preventative strategy is essential. The sensitivity of the anthropometric assessment instrument should be investigated for future studies. Therefore, it is preferable to investigate the sensitivity and its relationship to the most promising biomarkers because they save healthcare costs and make obesity easier to detect early.

Our analysis reveals that individuals with overweight and obesity who undergo weight loss consistently exhibit alterations in their microbiota profile, resembling that of individuals with a healthy weight. These alterations include increased diversity and reduce intestinal permeability. However, it is important to note that there is considerable variation in the gut microbiota among individuals. Based on the present data, a reduction in calorie consumption is associated with many outcomes, including weight loss, enhanced variety of gut microbiota, and a decrease in bacterial byproducts such as lipopolysaccharides. These modifications may lead to an increase in tight junction cohesiveness, a decrease in intestinal permeability, a reduction in liver exposure to these metabolites, and a suppression of pro-inflammatory pathways. To establish a compelling causal relationship, it is necessary to conduct randomized controlled trials with larger sample sizes and longer follow-up periods. Additional investigation is required in order to ascertain the specific taxonomic alterations at the phylum, genus, and species levels that contribute to increased biodiversity. Given that the majority of the aggregated estimates at these levels were derived from a limited number of studies, it is plausible that comprehensive documentation of microbiota alterations was insufficient, thereby explaining the observed evidence of overall diversity modifications but the absence of evidence regarding changes in the majority of phyla, genera, and species.

## Figures and Tables

**Figure 1 jpm-13-01602-f001:**
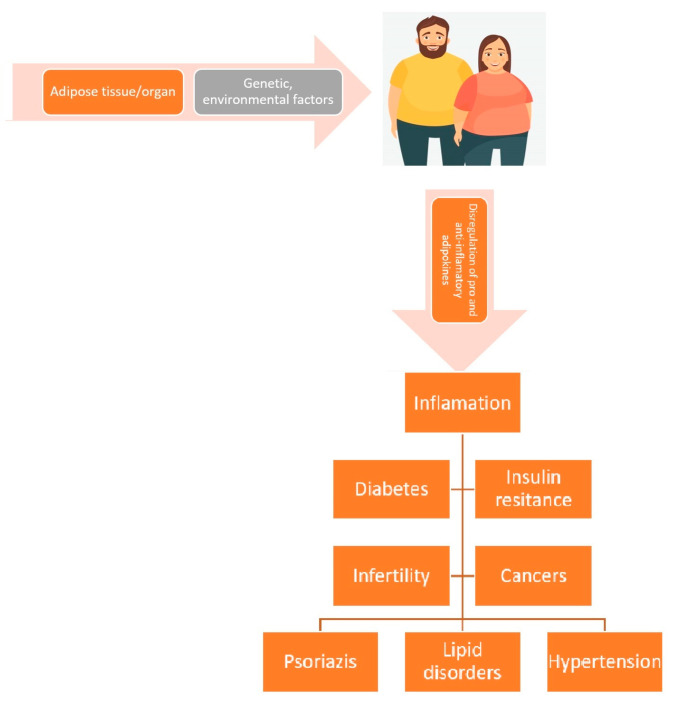
As obesity increases, the adipokine dysregulation pathway may help to cause long-term effects including metabolic disorders. Adapted from Zorena et al. [29].

**Figure 2 jpm-13-01602-f002:**
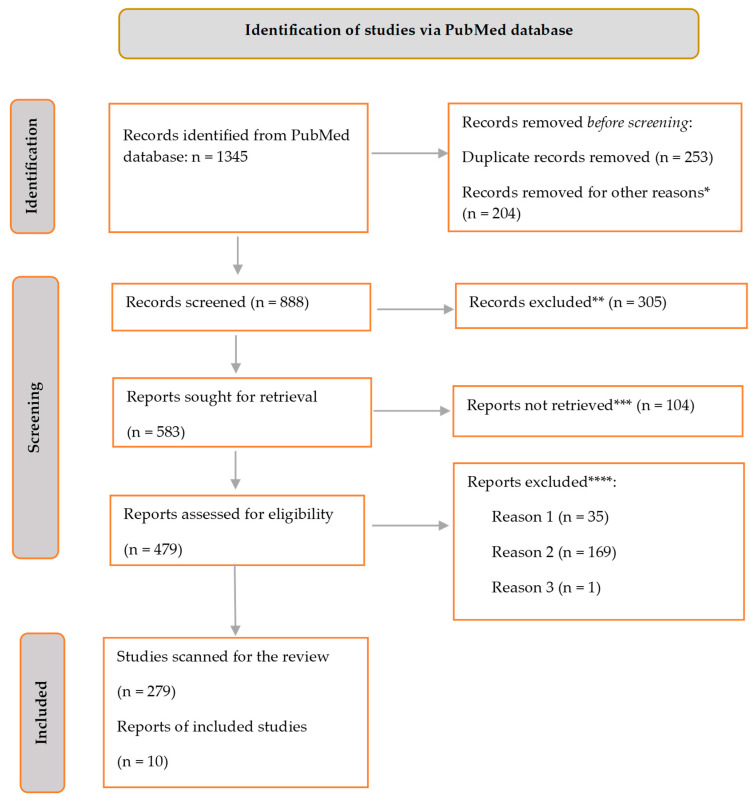
Flowchart depicting the process of document selection in the systematic literature review, according to the PRISMA paradigm. * The studies under consideration are not relevant to the current review. ** The studies do not contribute to our ability to address the research issue. *** The whole text of the study could not be located. **** Reason 1: the availability of free full text is restricted, therefore limiting access to comprehensive information. Reason 2: the study focuses on an age range that is not aligned with the intended target population. Reason 3: study was not written in the English language.

**Figure 3 jpm-13-01602-f003:**
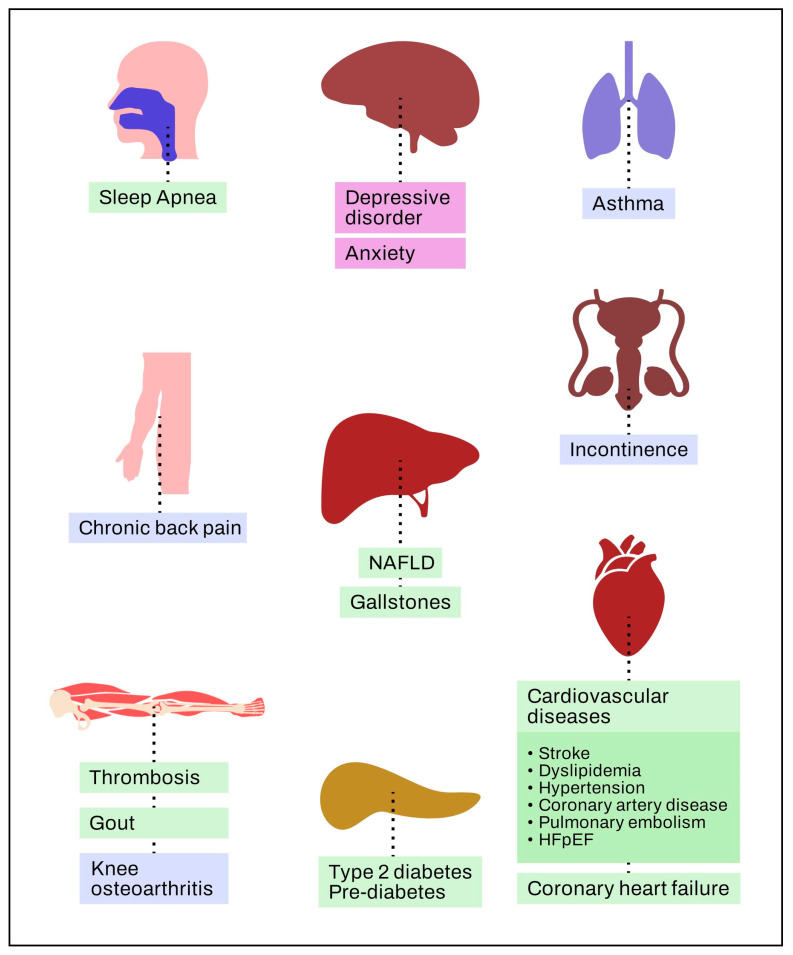
Obesity and its problems. Adapted from Sharma et al. [46].

**Figure 4 jpm-13-01602-f004:**
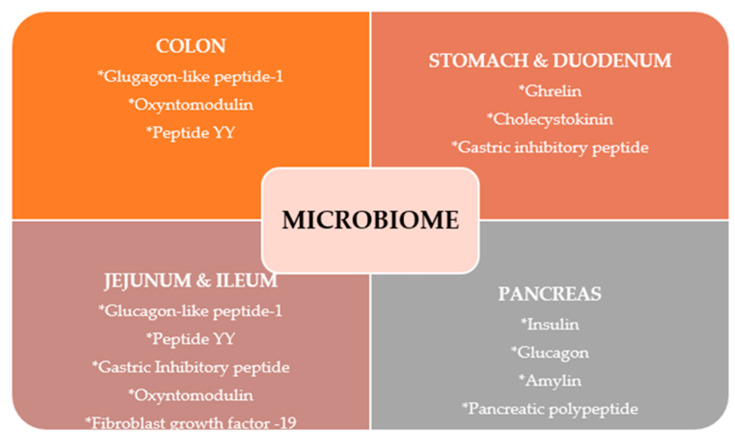
Hormones released by GM. Adapted from Côté et al. [50].

**Figure 5 jpm-13-01602-f005:**
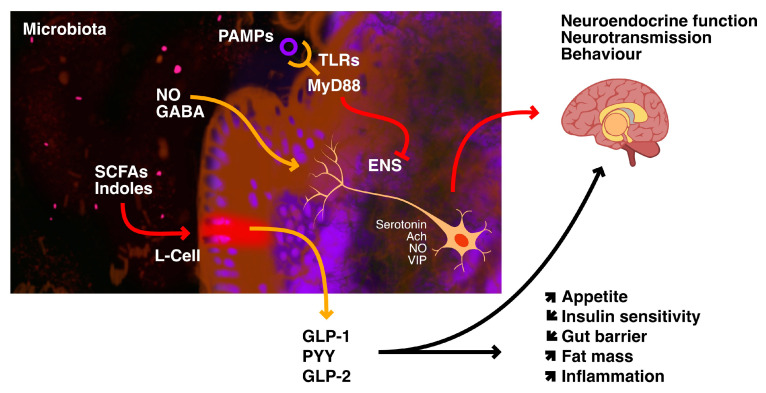
A description of the many interactions between microbial metabolites, endocrine systems, and neurological systems. Adapted from Cani et al. [56].

**Figure 6 jpm-13-01602-f006:**
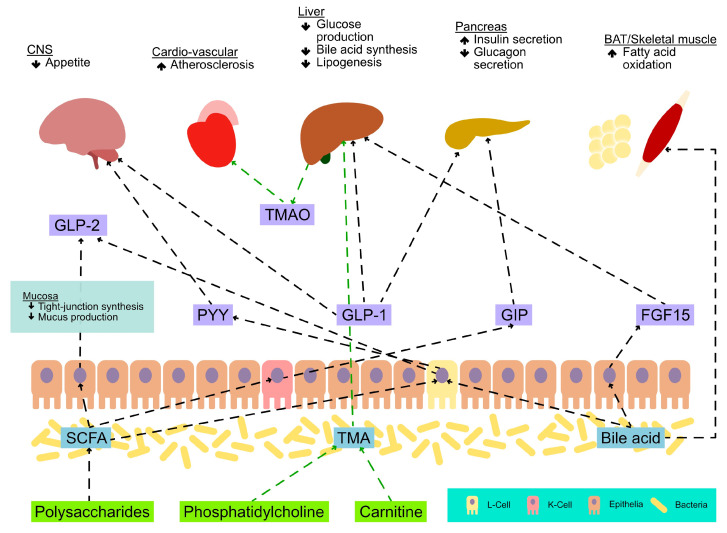
Gut microbiota–host interactions. Adapted from Hansen et al [59].

**Figure 7 jpm-13-01602-f007:**
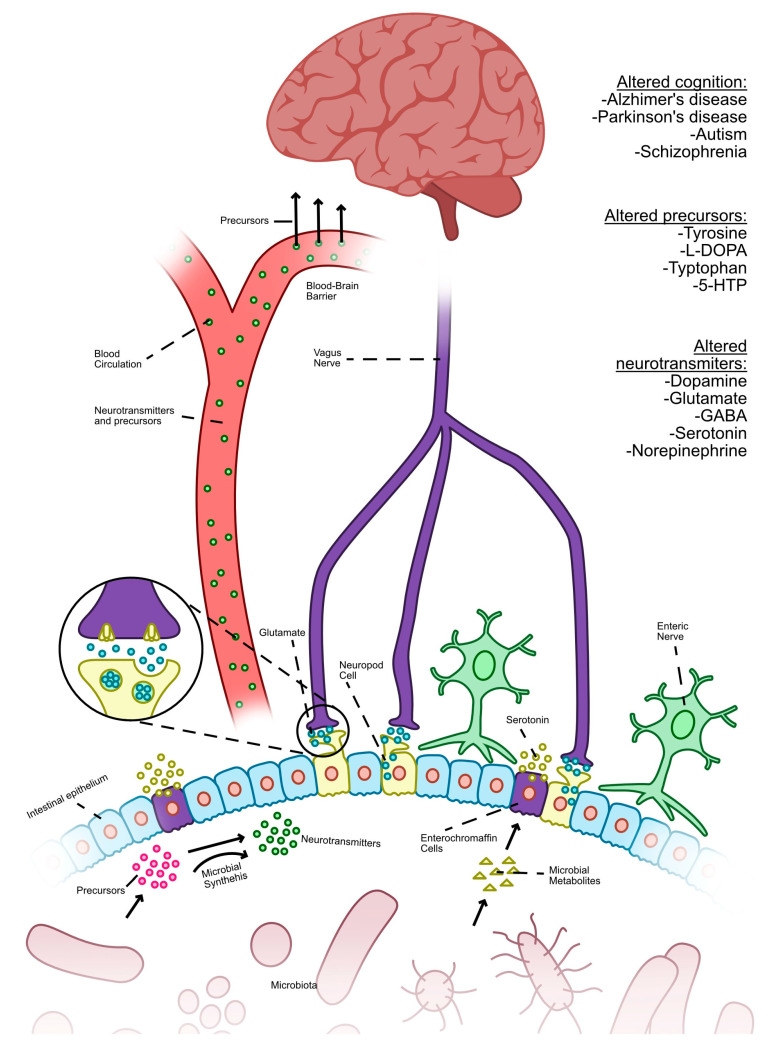
The impact of gut microbiota on the production of neurotransmitters and its influence on cognitive function (Adapted from [63]).

**Table 1 jpm-13-01602-t001:** Global overweight and obesity 2020–2035. World Obesity Atlas 2023. (n.d.). Retrieved 5 July 2023. Available online: https://www.worldobesity.org/resources/resource-library/world-obesity-atlas-2023 (accessed on 5 July 2023) (Adapted from World Obesity Atlas 2023).

	2020	2025	2030	2035
Number with overweight or obesity (BMI ≥ 25 kg/m^2^) (millions)	2603	3041	3507	4005
Number with obesity (BMI ≥ 30 kg/m^2^) (millions)	988	1249	1556	1914
Proportion of the population with overweight or obesity (BMI ≥ 25 kg/m^2^)	38%	42%	46%	51%
Proportion of the population with obesity (BMI ≥ 30 kg/m^2^)	14%	17%	20%	24%

**Table 2 jpm-13-01602-t002:** Results of the selected studies.

Study	Study Design	Pico Framework	Results of the Study	Conclusions	Links—MD *, MB **, and Obesity
Heianza et al. [36]	RCT	Population: 583 patients with G allele as Bifidobacterium-abundance-increasing allele. Intervention: The subjects were assigned at random to 1 of 4 diets for weight loss that varied in their macronutrient composition. Comparison: The study assessed adiposity measures over a span of two years, examining the correlation between the LCT genotype and weight-loss interventions. Outcomes: To see if there is a connection between the gut microbiota and obesity.	The researchers observed that alterations in overall body fat percentage, abdominal fat percentage, superficial adipose tissue mass, visceral adipose tissue mass, and total adipose tissue mass were markedly impacted by the LCT genotype and dietary protein consumption. The study found that those who had the G allele of the LCT variation rs4988235 saw a more significant decrease in many measures of body fat, including whole-body total percentage of fat, abdominal fat, superficial adipose tissue, visceral adipose tissue, and total adipose tissue, in response to a high-protein diet. In contrast, the G allele is often linked to less mitigation of these consequences when exposed to a low-protein dietary regimen.	The influence of the Bifidobacterium-related LCT genotype and dietary protein intake on the long-term enhancement of body fat composition and distribution was shown to be significant. The implementation of a dietary regimen that is both low in calories and rich in protein has the potential to assist individuals who are classified as obese or overweight, particularly those who possess the G allele of the LCT variant rs4988235, in decreasing their adiposity.	YES
Cuevas-Sierra et al. [37]	RCT	Population: Two hypocaloric diets were given a random assignment to 190 overweight and obese Spanish individuals for a period of four months. Intervention: A diet with moderately high protein was followed by 61 women and 29 men, and a diet with low fat was followed by 72 women and 28 men. Comparison: Four microbiota subscores related to the proportion of BMI loss for each diet were created using baseline fecal DNA, which was sequenced. Outcomes: To see if there is a connection between the gut microbiota and obesity.	The groups who used the MHP diet showed a large rise in protein consumption and a moderately significant drop in fat consumption, whereas the LF-diet group showed an increase in carbohydrate consumption and a considerable decrease in fat consumption. Women showed significantly reduced values for hip circumference after the MHP diet and lower values for leptin after the LF diet, but significant increases in HDL cholesterol. Following the LF diet, men showed considerably bigger reductions in weight, waist circumference, LDL cholesterol, and triglycerides, but a greater reduction in adiponectin levels with the MHP diet.	Despite having lower baseline values than women, men experienced a greater decline in adiponectin. According to the diet suggested for each group, the proportion of macronutrient intake showed substantial alterations as expected. As a result, a most effective weight loss plan based on this model was significantly assigned to a total of 72% of women and 84% of men who took part in this study.	YES
Leyrolle et al. [38]	RCT	Population: 106 obese patients. Intervention: Patients were assigned to two groups: prebiotic vs. placebo. Comparison: In addition to dietary guidance to consume inulin-rich or -poor vegetables for three months and to limit calorie consumption, patients received 16 g per day of native inulin or maltodextrin. Outcomes: To assess if there is a link between microbiota and obesity.	Except for inhibition, which was better in the prebiotic group, baseline mood and cognitive metrics did not change between the prebiotic and placebo groups. The placebo group had considerably more alcohol consumption at baseline. The therapy differently altered emotional competence. Even though within-group comparisons were not significant, emotional competence does, in fact, tend to rise in the prebiotic group while falling in the placebo group. Only in the prebiotic group did within-group comparisons show a significant reduction in negative feeling as judged by the Scale of Positive and Negative Experience and better flexibility.	Overall, the conclusion is that gut microbiota could be used to forecast how a prebiotic strategy will affect obese participants’ mood. It will be easier to tailor these methods if key gut bacteria in the body’s reaction to food-based therapy are identified. According to this research, Coprococcus may have neuroactive qualities and can be used as a gut microbiota biomarker for reaction to prebiotics.	YES
Zeng et al. [39]	RCT	Population: 1914 individuals average 41 years old, representing four typical lifestyles and living conditions in China. Intervention: Males made up 58%, and 11% of the total were healthy adults with normal BMIs and body weights. Comparison: Depending on the results of their physical examination and body mass index, the participants were divided into three groups: a healthy group, an obesity group without metabolic abnormalities, and an obesity group with abnormal clinical indications. Outcomes: To assess if there is a link between microbiota and obesity.	Patients with metabolic disorders showed changed GM components in comparison to obese patients without abnormalities, and Clostridium XIVa helped distinguish between obese patients with high serum cholesterol or blood pressure. These indicators revealed common GM changes in obese patients with various metabolic disorders, suggesting that other variables (such as genetic variation) may play a role in the development of several metabolic diseases. Based on these findings, the authors hypothesized that MDs were first brought on by obesity-related GM changes, and that further specific pathogenic aspects beyond GM dysbiosis needed to be investigated.	As a result, the study provided markers for obese individuals with diverse MDs, identified GM characteristics, and demonstrated the relationships between bacterial commensals and other clinical indications. These findings provided prospective GM targets for adjuvant therapies in the treatment of obesity with metabolic abnormalities and revealed the roles of GM in the etiology of metabolic disorders.	YES
Ghusn et al. [40]	RCT	Population: A total of 175 patients with BMI of 27 or more. Intervention: The patients were given weekly subcutaneous injections of semaglutide for a duration of at least three months. Comparison: A total of 132 female individuals were included in the study at the 3-month mark, whereas the number of patients decreased to 102 at the 6-month mark. Outcomes: To assess any connections of microbiota with obesity.	Following a period of three months, the mean reduction in weight was seen to be 6.7 kg, equivalent to 5.9% of an individual’s initial body weight. Subsequently, following a span of six months, the average weight loss increased to 12.3 kg, corresponding to 10.9% of one’s initial body weight. Among the cohort of 102 individuals who were subjected to monitoring over a period of 6 months, it was seen that a substantial proportion, namely 87.3%, had achieved a reduction in their body weight of no less than 5%. Furthermore, a significant percentage of 54.9% had managed to lose at least 10% of their initial body weight. Additionally, a noteworthy subset of 23.5% had successfully achieved a weight loss of at least 15%, while a smaller fraction of 7.8% had accomplished a reduction of no less than 20% of their initial body weight. At the 3-month and 6-month marks, those diagnosed with type 2 diabetes had comparatively lower average weight loss in comparison to those without the illness. Specifically, the weight loss percentages were 3.9% and 6.3% at 3 months and 7.2% and 11.8% at 6 months, respectively.	The results of this cohort study suggest that the weight reduction achieved with weekly dosages of 1.7 mg and 2.4 mg of semaglutide is similar to that found in randomized clinical trials.	YES
Zhou et al. [41]	RCT	Population: 264 overweight and obese patients. Intervention: From the beginning of the dietary intervention to six months later, blood levels of TMAO, choline, and l-carnitine were measured. Comparison: There were four different diets: two low fat, two high fat, two intermediate protein, and two high proteins. Outcomes: To find out if variations in BMD after two years were related to variations in plasma TMAO, choline, and l-carnitine levels from baseline to six months.	The researchers discovered that a higher loss in bone mineral density (BMD) at 6 months and 2 years was connected to a greater decline in plasma levels of TMAO from baseline to 6 months. Independent of changes in body weight, the larger decline in TMAO was also linked to a bigger loss in spine BMD at 2 years. The correlations were unaffected by the glycemic and diabetic status at baseline. In relation to changes in spine BMD and hip BMD after 6 months, L-carnitine alterations showed interactions with dietary fat intake. In the low-fat-diet group, those who saw the least drop in L-carnitine experienced less bone loss than in the high-fat-diet group.	Independent of diet treatments with different macronutrient contents and baseline diabetes risk factors, TMAO may protect against BMD decline during weight loss. The relationship between changes in plasma L-carnitine levels and changes in BMD may be altered by dietary fat. The results emphasize the significance of researching the link between TMAO and bone health in diabetic patients.	YES
Christensen et al. [42]	RCT	Population: 2224 individuals (1504 women and 720 men). Intervention: participants followed a low energy diet (LED) for 2 months. Comparison: Phase 1 consisted of an eight-week weight-loss phase using the LED. Phase 2 was a 148-week ongoing randomized lifestyle intervention that emphasized nutrition and exercise. Outcomes: To evaluate behavior modification for weight loss maintenance.	Men lost more weight than women (11.8% vs. 10.3%, respectively), but improvements in insulin resistance were comparable in both sexes. Men experienced greater declines in heart rate, fibromyalgia, the Z-score for the metabolic syndrome, and the C-peptide, whereas women experienced greater declines in HDL cholesterol, free fat mass, hip circumference, and pulse pressure. A total of 35% of participants returned to normoglycemia after the LED.	Women and men experienced distinct outcomes from the 8-week low-energy diet. These findings, which point to gender-specific alterations following weight loss, are therapeutically significant. It is crucial to investigate whether rapid weight reduction in women causes higher declines in free fat mass, hip circumference, and HDL cholesterol, which could jeopardize long-term weight maintenance and cardiovascular health.	YES
Shank et al. [43]	RCT	Population: 103 adolescent girls reported losing control of their eating (LOC), thus gaining weight. Intervention: The girls underwent assessments for the metabolic syndrome at baseline and again six months later. Comparison: Participants were randomly assigned to either a 12-week interpersonal group psychotherapy program or a group health education control program. Outcomes: Considering baseline age, depressive symptoms, fat mass, and height, the primary impacts of LOC status at treatment’s conclusion on metabolic syndrome components at a 6-month follow-up were studied.	Adolescents who had loss of control (LOC) remission at the conclusion of their therapy exhibited decreased levels of triglycerides, increased levels of high-density lipoprotein (HDL), and decreased levels of low-density lipoprotein (LDL) at the 6-month follow-up, in contrast to adolescents who continued to have persistent LOC. Notably, there were no discernible variations in these lipid components at baseline between the two groups. There were no significant differences seen in any other component based on the eating status of individuals with limited or no control (LOC) over their eating behavior.	Improvements in various metabolic syndrome components are linked to LOC eating remission. Future studies should continue to clarify the connection between LOC eating and physical health to conclude whether metabolic health may be improved in the long run by abstinence from LOC eating. The consumption of low-quality, energy-dense foods is a potentially modifiable lifestyle factor that might be strategically addressed in order to mitigate the risk of developing total or partial metabolic syndrome, provided that it leads to sustained improvements in metabolic well-being.	YES
Kwee et al. [44]	RCT	Population: 2458 participants were studied from 2006 and 2009 to 31 January 2015. Intervention: Using mass-spectrometry-based techniques, the quantitative levels of 135 metabolites were assessed at baseline. Comparison: The results were compared to see which group managed to have a diabetes remission status. Outcomes: To assess the change in diabetes-related clinical variables from pre-intervention to two years after-intervention.	Two metabolite factors, one with betaine and choline and the other with branched chain amino acids and tyrosine, were linked to the remission of diabetes.	The circulating baseline biomarkers for diabetes remission that the authors identified have independent associations as well as incremental predictive powers when included in a clinical model.	YES
Pearl et al. [45]	RCT	Population: 178 obese adults signed up for a weight-loss trial. Intervention: The participants filled in the Weight Bias Internalization Scale (WBI) and Patient Health Questionnaire. Comparison: The adults were investigated to determine whether WBI and metabolic syndrome (MS) are related. Outcomes: If participants lost less than 5% of their starting weight during a 14-week diet run-in period, they were randomly allocated to a 1-year weight reduction maintenance program to examine its effects.	Participants with higher WBI had an increased chance of fulfilling the criteria for MS. Greater likelihood of having high triglycerides were indicated by higher WBI. When categorically analyzed, high (vs. low) WBI indicated a higher likelihood of metabolic syndrome and high triglycerides.	Self-stigmatizing obese people may be at higher risk for cardiovascular and metabolic problems. Further exploring the biological and behavioral processes that connect WBI and metabolic syndrome is important.	YES

* MD—metabolic disorders; ** MB—microbome.

**Table 3 jpm-13-01602-t003:** Chen et al. [63]—synthesis of neurotransmitters.

Neuro- Transmitters	Precursor	Gut Microbiota	Cells of Intestine	Gut–Brain Axis
Glutamate (GLU)	Acetate	*Lactobacillus plantarum* *Bacteroides vulgatus* *Campylobacter jejuni*	Enteroendocrine cells	The transmission of sensory information originating from the intestines to the brain occurs through the vagus nerve.
GABA	Acetate	*Bifidobacterium* *Bacteroides fragilis* *Parabacteroides* *Eubacterium*	Myenteric neurons Mucosal endocrine-like cells	This neurotransmitter modulates the neuro-synaptic transmission in the GI nervous system and has an impact on intestinal motility and inflammation.
Acetylcholine	Choline	*Lactobacillus plantarum* *Bacillus acetylcholini* *Bacillus subtilis* *Escherichia coli* *Staphylococcus aureus*	Myenteric neurons	The myenteric neurons in the human colon are responsible for the production of 33% of the total output. The regulation of intestinal motility, secretion, and enteric neurotransmission is of paramount importance in maintaining proper gastrointestinal function.
Dopamine	Tyrosine l-DOPA	*Staphylococcus*		Affects gastric secretion, motility, and mucosal blood flow. Affects gastric tone and motility through nigro–vagal pathway in a Parkinson’s disease rat model.
Serotonin	5-HTP Tryptophan	*Staphylococcus* *Clostridial species*	Enterochromaffin cells	Enhance gastrointestinal peristalsis.
Norepinephrine	Tyrosine			Modulates energy intake and thermal homeostasis.
Tyramine	Tyrosine	*Staphylococcus* *Providencia*		The substance that precedes or serves as a precursor to octopamine.
Phenyle- thylamine	Phenyl- alanine	*Staphylococcus*		
Tryptamine	Tryptophan	*Staphylococcus* *Ruminococcus gnavus* *Clostridium sporogenes*		The stimulation of serotonin secretion in enterochromaffin cells. Enhances gastrointestinal function.

## Data Availability

Supplementary data can be provided on request.

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
