# Peer review of "Metabolic Disorders, the Microbiome as an Endocrine Organ, and Their Relations with Obesity: A Literature Review"

_jpm, 2023, doi:10.3390/jpm13111602_

Round 1

Reviewer 1 Report

Comments and Suggestions for Authors

I read with interest the paper by Ispas et al, concerning the role of microbiota in obesity. The paper is interesting, however I have some suggestions.

The re are some typing errors, in particular "addiposity" in fig 1. The authors moreover should check the manuscript, since many times  the term microbioma is used instead of microbiota

Comments on the Quality of English Language

Not bad, there are some typing errors

Reviewer 2 Report

Comments and Suggestions for Authors

The authors have wirtten a very detailed and comprehensive review . The mansucript was compiled and evaluated from a myriad  of different studies on the topic. Data on the microbiome are extremely numerous and it is helpful to have a comprehensive overview of the topic. In this respect the authors have condcted a valuable piece of work, which will be of interest for a great number of readers.
